# Unhygienic stool-disposal practices among mothers of children under five in Cambodia: Evidence from a demographic and health survey

**Pisey Vong**[1]*, **Pannee Banchonhattakit**[2], **Samphors Sim**[3], **Chamroen Pall**[4], **Rebecca S. Dewey**[5]

**1** Office of Rural Health Care, Provincial Department of Rural Development, Pursat, Cambodia, **2** Faculty of Public Health, Khon Kaen University, Khon Kaen, Thailand, **3** Chea Sim University of Kamchaymear, Prey Veng, Cambodia, **4** Ministry of Education, Youth and Sport, Phnom Penh, Cambodia, **5** University of Nottingham, Nottingham, United Kingdom

* piseyvong11@gmail.com

**Data Availability Statement:** Data can be downloaded from the DHS Data Archive after user registration and getting approval for dataset

## Abstract

### Background

Unhygienic disposal of children's stools affects children's health in terms of their susceptibility to many diseases. However, there are no existing studies into the impact of unhygienic stool disposal in Cambodia. Therefore, this study aimed to identify factors associated with the unhygienic disposal of children's stools among children under the age of five in Cambodia.

### Methods

An analytical cross-sectional study was conducted using data from the Cambodia Demographic and Health Survey (CDHS) 2014. A multivariable binary logistic regression was conducted using Stata to analyze factors associated with the unhygienic disposal of children's stools.

### Results

Overall, the prevalence of practicing unhygienic disposal of children's stools was 29.27% (95%CI: 27.51%- 31.09%). Factors statistically associated with this practice were: living in the Central Plain, Plateau and Mountains, Coastal and Sea regions (AOR = 1.65; 95% CI: 1.33–2.04), (AOR = 2.53; 95% CI: 1.98–3.24) and (AOR = 4.16; 95% CI: 3.15–5.48) respectively, poor household wealth (AOR = 1.58; 95% CI: 1.31–1.91), the mother having no education (AOR = 1.45; 95% CI: 1.14–1.85), a high number of children aged under five (AOR = 1.11; 95% CI: 1.03–1.20), being in the "other" religious category (AOR = 1.77; 95% CI: 1.25–2.51), living in a household with unimproved toilet facilities (AOR = 1.22; 95% CI: 1.11–1.34), living in a household with inadequate hygiene (AOR = 1.33; 95% CI: 1.12–1.59), and the household not being visited by a family planning worker in the last year (AOR

access. The DHS Programme, Demographic and Health Surveys, Cambodia 2014: https:// dhsprogram.com/data/Access-Instructions.cfm.

**Funding:** The author(s) received no specific funding for this work.

**Competing interests:** The authors have declared that no competing interests exist.

= 1.45; 95% CI: 1.19–1.77). However, an increase in the child's age by even a month had significant negative associations with unhygienic practice (AOR = 0.65; 95% CI: 0.60–0.70), even when controlling for other covariates.

## Conclusion

Almost one third of the mothers do not practice hygienic disposal of children's stools in Cambodia. Unhygienic practices were more prevalent in certain regions, and were also associated with low wealth, lack of education, an increase in the number of children under five in the household, religion, lack of sanitation and access to healthcare professionals. Conversely, the child's age was found to be positively associated with the hygienic disposal of children's stools.

## Introduction

Globally, 2.4 billion people have no access to improved sanitation facilities. One in eight people worldwide (a total of 946 million people) practiced open defecation (OD) in 2015 [1], which decreased to 673 million in 2017 [2]. OD can cause both acute and chronic health effects [3]. Acute effects include infectious intestinal diseases including diarrheal diseases, adverse pregnancy outcomes, and life-threatening violence against women and girls. Chronic effects include soil-transmitted helminthiases, increased anemia, giardiasis, environmental enteropathy and small- intestine bacterial overgrowth, and stunting and long-term impaired cognition [3]. In 2015, 47% of the population in Cambodia practiced OD [4]. According to the WHO/ UNICEF Joint Monitoring Programme (JMP) for Water Supply and Sanitation, a child's feces is classified as being disposed of hygienically when the child passes stools using a toilet or latrine, the feces are put or rinsed into the toilet or latrine, or the feces are buried [5]. Unsanitary disposal of children's feces can impact children's health in ways such as environmental enteropathy, impaired growth, and childhood gastroenteritis [6].

Cambodia has made significant progress in the provision of improved sanitation facilities, marked by an increase in availability from 3% to 42% between 1990 and 2015. Conversely, this decreased the practice of OD from 89% to 47% in the same period [4]. Cambodia's Ministry of Rural Development, (MRD); in collaboration with partners and Non-Governmental Organizations (NGOs) has been implementing the Community-Led Total Sanitation (CLTS) program according to the national guidelines. CLTS has been active in Cambodia since 2005. CLTS is an innovative approach for mobilizing communities to completely eliminate OD. CLTS focuses on the behavioral change needed to ensure real and sustainable improvements [7].

CLTS focuses on hygienic behaviors including the safe disposal of a child's feces, as well as stopping OD, ensuring hygienic toilet use, hygienic practices around food, washing hands at appropriate times; and maintaining a clean and safe environment [8]. The Royal Government of Cambodia (RGC) aims to achieve 100% availability of Water and Sanitation by 2025, clearly stating that "everyone must have access to a water supply and live in a hygienic environment by 2025" [9]. Therefore, unhygienic disposal children's feces is one of the most important problems to solve.

It is of great importance to understand the factors associated with unhygienic stool disposal practices affecting children under the age of five in Cambodia. At the time of writing, there are no national studies on unhygienic practices around the disposal of children's stools in Cambodia.

## Methods

### Data source

This research study was an analysis of secondary data from the Cambodia Demographic and Health Survey (CDHS), 2014. The CDHS 2014 is the fourth CDHS targeted to provide information related to fertility, family planning, maternal and child health, gender, HIV/AIDS, malaria, and nutrition. Out of the seven data files available, two relevant data files were included in the study; one containing household data and the other containing children's data. Further detailed information about the sampling methods used and about CDHS can be found in the CDHS report [10].

### Inclusion/exclusion criteria

Data pertaining to practices related to the disposal of children's stools were collected for all children born during the five years preceding the 2014 CDHS. If more than one child aged under five was living in a household with their mother, we included only the youngest child in each household in our analysis.

### Outcome variable

The prevalence of unsanitary disposal of children's stools is the outcome variable considered in this study. It is referred to in the questionnaire thus: "The last time (NAME FROM 553) passed stools, what was done to dispose of the stools?" According to the WHO/UNICEF definition [5], we generated and recoded a new binary variable: the child using a toilet or latrine, disposing of or rinsing the child's stool into a toilet or latrine, or burying the child's stool were all coded as hygienic disposal of children's stool and assigned the value "0". Conversely, disposing of or rinsing the child's stool into a drain or ditch, throwing it into garbage, leaving it in the open or not disposing of at all were assigned the code "1".

### Explanatory variables

Explanatory variables used were geographical region, residence, wealth index, maternal and paternal education, child's age, mother's age, number of children aged 5 years and under in the household, religion, parental marital status, drinking water source, and whether or not the same source was used during dry and wet seasons, type of toilet facility, use of hygiene, exposure to media, and access to family planning and health professionals. Geographical regions were [11] *Tonle Sap*; *Central Plain*; *Plateau and Mountains*; and *Coastal and Sea* (Fig 1). The residence was classified as either *urban* or *rural*. Wealth index was recoded as a binary variable, with those in the poorest and poorer quintiles both coded as *poor*, and those in middle, richer and richest quintiles coded as *non-poor* [12]. Mother's and father's education were recoded as separate binary variables, with *no education* retaining the classification, and those receiving primary, secondary and higher levels of education were all recoded as having had *education* [13]. The *child's age* was treated as a continuous variable. The *mother's age* was coded as a discrete variable with three coded values; *15–24 years*, *25–34 years* and *≥ 35 years*). The *number of children aged 5 years and under* in the household was treated as a continuous variable. Religion was recoded as binary variable, *Buddhist* and *non-Buddhist* (Muslim, Christian, and other religions were recoded as other religion). Maternal marital status was recoded as a binary variable; married and living with partner were coded as *living with partner*, whereas widowed, divorced, and no longer living together/separated were recoded as *separated*. The source of drinking water used by the household during the dry season was recoded as either *improved* or *unimproved* [14]. Similarly, the source of drinking water during wet season was recoded as

| Tonle Sap | Banteay Meanchey, Battambang, Kampong Chhnang, Kampong Thom, Pursat, Siem Reap, Otdar Meanchey, and Pailin |
| --- | --- |
| Central Plain | Kampong Cham, Tbong Khmum, Kandal, Phnom Penh, Prey Veng, Svay Rieng, and Takeo |
| Plateau and Mountains | Kampong Speu, Kratie, Mondul Kiri, Preah Vihear, Ratanak Kiri, and Stung Treng |
| Coastal and Sea | Kampot, Koh Kong, Preah Sihanouk, and Kep |

**Fig 1. Geographical regions.**

either *improved* or *unimproved* [14]. Use of the same or different sources of drinking water during both wet and dry seasons was coded as either *same* or *different*. The type of toilet facility used by members of the household was recoded as either *improved* or *unimproved* [14]. Whether or not adequate hygiene was used in the household was recoded as a binary variable; *using adequate hygiene* and *not using adequate hygiene* [14]. Exposure to media was recoded as a binary variable; *exposed* and *not exposed* [15]. The occurrence of visits by family planning health workers in the last year was coded as either *yes* or *no*. Similarly, whether or not they had visited a health institution in the last year was coded as *yes* or *no*. The children's experience of diarrhea in the last two weeks was coded as a binary variable; those responding no and don't know were recoded as *no*, and those responding yes retained the coding of *yes*.

## Data use

Data from the CDHS 2014 were used in this study.

## Statistical analysis

The analysis was performed using STATA/SE version 14.0 [16]. The women's individual sample weightings were used in the estimation to provide nationally representative results [17]. Frequency analysis of categorical data was performed to provide number and percentage. Continuous data were analysed as means, standard deviations, and ranges. Bivariate and multivariable binary analyses were used to assess the effects and interactions of associated factors on the unhygienic disposal of children's stools. A linearity test was conducted between every continuous explanatory variables and outcome variable. Any explanatory variables significant at $p<0.25$ in the unadjusted binary logistic regression were entered into the initial model [18, 19]. Multi-collinearity of the explanatory variables was assessed by excluding those with a variance inflation factor (VIF) greater than 4 and no variable with VIF greater than 4 was found.

Multivariable binary logistic regression was used to find the factors associated with the unhygienic disposal of children's stools. The *svyset* command was used to test for complex survey sampling methods used in the original surveys, in order to adjust for differences in the probabilities of sample selection, and to avoid using over-sampled strata within the survey data [20]. Differences associated with factors are reported using the adjusted odds ratio (AOR).

### Ethics

Data from the CDHS were used after having obtained written permission from the ICF with the ref. number 125453. As part of the first author's PhD at Khon Kaen University, this study was conducted following additional approval received from the Khon Kean University Ethics Committee in Human Research with the ref. number HE632097.

## Results

Among the 5,745 households in the sample, almost half lived in the Central Plain region. The majority lived in rural areas. Almost half were classified as poor wealth. The majority of parents had received education. The average child age was 26.16±16.94 months, and the average maternal age was 28.76±6.10 years. The majority of the sample were Buddhist. Almost two thirds of the sample used an improved drinking water supply during the dry season while the majority used an improved drinking water supply during the wet season. Less than half of the sample used improved toilet facilities and about two thirds used adequate hygiene. Less than one third had been visited by a family planning health worker in the last year, whereas more than half had visited a healthcare facility in the last year. About thirteen percent of children had experienced diarrhea in the last two weeks (Table 1).

### Factors associated with unhygienic stool disposal among mothers of children under five in Cambodia using simple logistic regression

Table 2 shows the factors that had statistically significant associations with unhygienic stool disposal among children under five (p<0.05); these were region, wealth index, maternal and paternal education, the child's age, the mother's age, the number of children in the household aged under five, the religion, the source of drinking water during dry season, whether or not the same source of drinking water was used during wet and dry season, the toilet facility, use of hygiene, exposure to the media, whether or not the family had been visited by a family planning health worker in the last year, and whether or not the child had experienced diarrhea in the last two weeks. The remaining factors of the source of drinking water used during the wet season, and whether or not the family had visited a healthcare institution in the last year did not reach significance, however they did reach the pre-determined threshold of p<0.25 for inclusion in the initial model.

### Factors associated with unhygienic stool disposal among mothers of children under five years in Cambodia using multiple logistic regression

Table 3 shows that households in the Central Plain region, Plateau and Mountains region, and Coastal and Sea region had higher odds ([AOR = 1.65; 95% CI: 1.33–2.04], [AOR = 2.53; 95% CI: 1.98–3.24], and [AOR = 4.16; 95% CI: 3.15–5.48], respectively)) of practicing the unhygienic disposal of children's stools than those in Tonle Sap region. The odds of practicing the unhygienic disposal of children's stools were 58% higher in households with poor wealth compared to non-poor and were 45% higher in mothers who had no education compared to those having education. As the child's age increased by a month, the odds of unhygienic stool

**Table 1. Characteristics of households in Cambodia, 2014 (n = 5,745).**

| Variables | Number | Percentage |
|---|---|---|
| **Region** | | |
| Tonle Sap | 1846 | 32.13 |
| Central Plain | 2605 | 45.34 |
| Plateau and Mountains | 933 | 16.24 |
| Coastal and Sea | 361 | 6.29 |
| **Residence** | | |
| Urban | 817 | 14.23 |
| Rural | 4928 | 85.77 |
| **Wealth index** | | |
| Non-poor | 3255 | 56.65 |
| Poor | 2490 | 43.35 |
| **Maternal Education** | | |
| Had education | 4974 | 86.58 |
| No education | 771 | 13.42 |
| **Paternal Education** | | |
| Had education | 5207 | 90.62 |
| No education | 538 | 9.38 |
| **Child's age (months)** | | |
| 48–59 | 863 | 15.04 |
| 36–47 | 940 | 16.37 |
| 24–35 | 1136 | 19.77 |
| 12–23 | 1368 | 23.80 |
| <12 | 1438 | 25.03 |
| Mean±SD | 26.16±16.94 | |
| Range | 0 to 59 | |
| **Mother's age (years)** | | |
| ≥35 | 925 | 16.11 |
| 25–34 | 3232 | 56.25 |
| 15–24 | 1588 | 27.64 |
| Mean±SD | 28.76±6.10 | |
| Range | 15 to 49 | |
| **Number of children under five years** | | |
| 1 | 3812 | 66.35 |
| 2–6 | 1933 | 33.65 |
| Mean±SD | 1.41±0.64 | |
| Range | 1 to 6 | |
| **Religion** | | |
| Buddhism | 5495 | 95.64 |
| Other | 250 | 4.36 |
| **Mother's marital status** | | |
| Separated | 263 | 4.58 |
| Living with partner | 5482 | 95.42 |
| **Source of drinking water during dry season** | | |
| Improved | 3566 | 62.07 |
| Unimproved | 2179 | 37.93 |
| **Source of drinking water during wet season** | | |
| Improved | 4737 | 82.45 |

(*Continued*)

**Table 1.** (Continued)

| Variables | Number | Percentage |
|---|---|---|
| Unimproved | 1008 | 17.55 |
| **Same source during wet and dry seasons** | | |
| Same | 3967 | 69.05 |
| Different | 1778 | 30.95 |
| **Toilet facility** | | |
| Improved | 2405 | 41.86 |
| Unimproved | 3340 | 58.14 |
| **Hygiene** | | |
| Adequate | 3845 | 66.93 |
| Inadequate | 1900 | 33.07 |
| **Exposure to media** | | |
| Yes | 3637 | 63.30 |
| No | 2108 | 36.70 |
| **Family planning health worker visit in last year** | | |
| Yes | 1473 | 25.65 |
| No | 4272 | 74.35 |
| **Visited healthcare institution in last year** | | |
| No | 2346 | 40.83 |
| Yes | 3399 | 59.17 |
| **Children with diarrhea in last two weeks** | | |
| No | 4968 | 86.46 |
| Yes | 777 | 13.54 |

disposal decreased by 35%. As the number of children under the age of five in the household increased by one, the odds of unhygienic stool disposal increased by 11%. The odds of practicing unhygienic disposal of children's stools was 77% higher in non-Buddhist households compared to Buddhist households. The odds were 22% higher in households with unimproved toilet facilities compared with those with improved toilet facilities. The odds were 33% higher in households using inadequate hygiene compared with those using adequate hygiene. The odds of practice unhygienic disposal of children's stools were 45% higher in households that had not been visited by a family planning health worker in the last year compared with those were visited.

## Discussion

Alarmingly, our study has shown that as many as one third of Cambodian mothers or caregivers of children practiced unhygienic stool disposal. However, this prevalence is not dissimilar to that found in a previous study conducted in Madagascar (38%) [21]. Furthermore, unhygienic practices are much less prevalent in Cambodia in comparison to Zambia, Kenya, Uganda and Malawi, with prevalence rates of 67%, 70%, 75%, and 79% respectively [22–25]. Conversely, other studies have found even higher prevalence rates (81.4%) in India and lower again in Ethiopia (38%) [26]. In our recent study, many associated factors have been found to be associated with unhygienic stool disposal practices: geographical region (with Central Plain region, Plateau and Mountains region, Coastal and Sea region being worst), poor household wealth, lack of maternal education, a high number of children under five children in the household, non-Buddhist religion, use of unimproved toilet facilities, adequate use of hygiene, and not being visited by a family planning worker in the last year. However, an increase in the

**Table 2. Unadjusted binary logistic regression of factors associated with unhygienic stool disposal (USD) of children under the age of five in Cambodia, 2014 (n = 5,745).**

| Variables | Number | USD % | odds ratio | 95% CI | p-value |
|---|---|---|---|---|---|
| **Overall** | 5745 | 29.27 | | 27.51–31.09 | |
| **Region** | | | | | <0.001 |
| Tonle Sap | 1846 | 21.61 | 1 | | |
| Central Plain | 2605 | 27.79 | 1.40 | 1.14–1.71 | |
| Plateau and Mountains | 933 | 41.53 | 2.58 | 2.01–3.30 | |
| Coastal and Sea | 361 | 47.35 | 3.26 | 2.46–4.34 | |
| **Residence** | | | | | 0.630 |
| Urban | 817 | 28.47 | 1 | | |
| Rural | 4928 | 29.40 | 1.05 | 0.87–1.26 | |
| **Wealth index** | | | | | <0.001 |
| Non-poor | 3255 | 22.78 | 1 | | |
| Poor | 2490 | 37.74 | 2.05 | 1.74–2.42 | |
| **Maternal Education** | | | | | <0.001 |
| Had education | 4974 | 27.57 | 1 | | |
| No education | 771 | 40.20 | 1.77 | 1.40–2.23 | |
| **Paternal Education** | | | | | <0.001 |
| Had education | 5207 | 28.31 | 1 | | |
| No education | 538 | 38.54 | 1.59 | 1.22–2.06 | |
| **Child's age (months)** | 5745 | N/A | 0.66 | 0.62–0.71 | <0.001 |
| **Mother's age (years)** | | | | | |
| ≥35 | 925 | 25.84 | 1 | | 0.005 |
| 25–34 | 3232 | 28.43 | 1.14 | 0.92–1.41 | |
| 15–24 | 1588 | 32.96 | 1.41 | 1.12–1.79 | |
| **Number of children under five years** | 5745 | N/A | 1.24 | 1.17–1.33 | <0.001 |
| **Religion** | | | | | <0.001 |
| Buddhism | 5495 | 28.33 | 1 | | |
| Other | 250 | 49.80 | 2.51 | 1.65–3.81 | |
| **Mother's marital status** | | | | | 0.252 |
| Separated | 263 | 25.56 | 1 | | |
| Live with partner | 5482 | 29.44 | 1.22 | 0.87–1.70 | |
| **Source of drinking water during dry season** | | | | | 0.003 |
| Improved | 3566 | 27.29 | 1 | | |
| Unimproved | 2179 | 32.49 | 1.28 | 1.08–1.52 | |
| **Source of drinking water during wet season** | | | | | 0.066 |
| Improved | 4737 | 28.48 | 1 | | |
| Unimproved | 1008 | 32.96 | 1.23 | 0.99–1.55 | |
| **Same source during wet and dry season** | | | | | 0.004 |
| Same | 3967 | 27.71 | 1 | | |
| Different | 1778 | 32.73 | 1.27 | 1.08–1.50 | |
| **Toilet facility** | | | | | <0.001 |
| Improved | 2405 | 21.25 | 1 | | |
| Unimproved | 3340 | 35.03 | 1.41 | 1.30–1.54 | |
| **Hygiene** | | | | | <0.001 |
| Adequate | 3845 | 25.81 | 1 | | |
| Inadequate | 1900 | 36.27 | 1.64 | 1.37–1.95 | |
| **Exposure to media** | | | | | <0.001 |

(*Continued*)

**Table 2.** (Continued)

| Variables | Number | USD % | odds ratio | 95% CI | p-value |
|---|---|---|---|---|---|
| Yes | 3637 | 26.06 | 1 | | |
| No | 2108 | 34.80 | 1.51 | 1.29–1.77 | |
| **Family planning health worker visit in last year** | | | | | 0.002 |
| Yes | 1473 | 25.14 | 1 | | |
| No | 4272 | 30.69 | 1.32 | 1.10–1.58 | |
| **Visited healthcare institution in the last year** | | | | | 0.051 |
| No | 2346 | 27.33 | 1 | | |
| Yes | 3399 | 30.60 | 1.17 | 0.99–1.37 | |
| **Children with diarrhea in the last two weeks** | | | | | <0.001 |
| No | 4968 | 28.16 | 1 | | |
| Yes | 777 | 36.36 | 1.46 | 1.18–1.80 | |

child's age by as little as a month had a significant negative associations with the unhygienic disposal of children's stools, when controlling for other covariates.

The region was significantly associated with unhygienic stool disposal, explicitly that the Coastal and Sea region, Central Plain, and Plateau and Mountains exhibited AORs 4.16 times, 2.53 times, and 1.65 times of that in the Tonle Sap lake region. This is in agreement with a study from Kenya which found that mothers and caregivers who are living in urban areas had

**Table 3. Multivariable analysis of factors associated with unhygienic stool disposal (USD) of children under the age of five in Cambodia, (n = 5,745).**

| Variables | Number | USD % | AOR | 95% CI | p-value |
|---|---|---|---|---|---|
| **Region** | | | | | <0.001 |
| Tonle Sap | 1846 | 21.61 | 1 | | |
| Central Plain | 2605 | 27.79 | 1.65 | 1.33–2.04 | |
| Plateau and Mountains | 933 | 41.53 | 2.53 | 1.98–3.24 | |
| Coastal and Sea | 361 | 47.35 | 4.16 | 3.15–5.48 | |
| **Wealth index** | | | | | <0.001 |
| Non-poor | 3255 | 22.78 | 1 | | |
| Poor | 2490 | 37.74 | 1.58 | 1.31–1.91 | |
| **Maternal Education** | | | | | 0.003 |
| Had education | 4974 | 27.57 | 1 | | |
| No education | 771 | 40.20 | 1.45 | 1.14–1.85 | |
| **Child's age (months)** | 5745 | N/A | 0.65 | 0.60–0.70 | <0.001 |
| **Number of children under five years** | 5745 | N/A | 1.11 | 1.03–1.20 | <0.001 |
| **Religion** | | | | | 0.001 |
| Buddhism | 5495 | 28.33 | 1 | | |
| Other | 250 | 49.80 | 1.77 | 1.25–2.51 | |
| **Toilet facility** | | | | | <0.001 |
| Improved | 2405 | 21.25 | 1 | | |
| Unimproved | 3340 | 35.03 | 1.22 | 1.11–1.34 | |
| **Hygiene** | | | | | <0.001 |
| Adequate | 3845 | 25.81 | 1 | | |
| Inadequate | 1900 | 36.27 | 1.33 | 1.12–1.59 | |
| **Family planning health worker visit in last year** | | | | | <0.001 |
| Yes | 1473 | 25.14 | 1 | | |
| No | 4272 | 30.69 | 1.45 | 1.19–1.77 | |

better understanding and awareness of sanitation and hygiene [27]. Therefore, households in the Coastal and Sea region are most at risk; it might be that the mothers and caregivers in this region have not received adequate information or intervention with regards to sanitation, due to the distance of the region from the city. Therefore, households in the rural areas of the region may be strongly impacted the practice of unhygienic stool disposal.

Wealth index was also significantly associated with unhygienic stool disposal, with mothers and caregivers living in poverty being 1.58 times (AOR 1.58, 95% CI 1.31–1.91) more likely to have unhygienic practices compared to those not in poverty. This finding is consistent with a study from India [28] which showed that the household earning ≤23.9 USD per month is less likely by a factor of 4.7 to have hygienic practices compared to those who earned >23.9 USD. Moreover, a study from Bangladesh [29] has shown that households with high socioeconomic status are less likely to practice unhygienic stool disposal compared to low socioeconomic status households [30]. It is likely that the mothers and caregivers in wealthier households experience a better standard of living and have more knowledge related to the hygienic disposal of children's stools. Further, a status of higher wealth will allow the mothers and caregivers to improve their standard of living.

The level of the mother's education plays an important role in the hygienic disposal of children's stools with 86% having received education. This study found that a mother who had no education was more likely (AOR 1.45; 95%CI: 1.14–1.85) to practice unhygienic stool disposal compared to a mother who had received formal education. This is in agreement with the findings of the National Family Health Survey (NFHS-3 2005–2006), in which unhygienic behaviors were 61% more likely in mothers who had finished secondary school compared to mothers who had not [31]. Similarly, a study from Accra found that mothers who had finished primary school were more likely to practice healthy stool disposal [32] than those who had not. Specifically, mothers who had awareness and understanding about the causes and impacts of unhygienic stool disposal were able to practice well compared to mothers who had no or a lower level of education [33]. It can be seen that education is absolutely vital for mothers and caregivers who are looking after children since they need knowledge about the root causes of problems and the impacts of their behaviors on the child's health.

An increase in the child's age by as little as a month had a significant, negative association with the practice of unhygienic disposal of children's stools. This is in line with a study conducted in India and Cambodia [13, 34] in which the older the child, the safer the practice of stool disposal. Additionally, our findings are that as the number of children under the age of five increased, the risk of unsafe disposal of the children's stool increased, again in line with a study in Ethiopia [35].

Importantly, the present study found that religion was significantly associated with unhygienic stool disposal practices. Mothers who were not Buddhist were 1.77 times more likely to practice unhygienic stool disposal compared to mothers who were Buddhist. This finding may be related to the practice in Buddhism of monks playing an important role during ceremonies in giving out advice to the people in the community, especially that related to hygiene and sanitation. Hence, followers of Buddhism have more time to engage in informal education from Buddhist monks.

Most importantly, the quality of the toilet facility is significantly associated with unhygienic stool disposal practices. Households that had improved toilet facilities were more likely to practice hygienic stool disposal (AOR: 1.22) compared to households where the toilet facility had not been improved. This result is in line with studies in India and rural Bangladesh [13, 36]. Moreover, this finding is consistent with studies reporting that the households failing to follow hygienic stool disposal practices had not improved their toilet [22, 37]. Similarly, in Ethiopia and South Africa, improvement of households' toilet facilities correlated highly with

the likelihood of practicing hygienic stool disposal [23, 25]. Therefore, sanitation interventions should not neglect the importance of behavioral changes. Latrine improvement is crucial to ensure that mothers and caregivers do not face additional risks resulting from their standard of living.

We also found inadequate use of hygiene generally was significantly associated with unhygienic stool disposal. Mothers who had inadequate hygiene were more likely (AOR = 1.33) to dispose of their child's stool unhygienically compared to mothers with adequate hygiene. This is consistent with a report from Ethiopia showing that mothers who used unimproved water sources were more likely to practice unhygienic stool disposal practices compared to mothers who favored piped water and water from other improved sources [35]. This finding further in line with the findings of a study conducted in Northern Ethiopia, which revealed that mothers who had used cleaning materials to wash their hands after using the toilet were more likely to practice unhygienic stool disposal compared to mothers using either water only or water and soap. It seems that unhygienic stool disposal is highly related to the mother's broader understanding of what good hygiene involves; further that as the knowledge and awareness of the mother improves, they are less likely to have unhealthy behaviors. Therefore, teaching the understanding of hygiene is paramount for improving quality of life.

Finally, not receiving a visit from a family planning health worker in the last year was significantly associated with unhygienic stool disposal. This finding is in agreement with those from the Ethiopian study [35] that found mothers who had not visited family planning health workers in the last year to be 1.50 times more likely to practice unhygienic stool disposal compared to mothers who had had contact with the family planning health worker in the last year. Receiving advice from family planning health workers is vital for wellbeing, especially for the mothers and caregivers who look after children because this advice provides be the awareness and knowledge necessary for disease prevention.

## Acknowledgments

The authors would like to express sincere thanks and appreciation to:

Dr. Kavin Thinkhamrop, Health and Epidemiology Geoinformatics Research (HEGER), Faculty of Public Health, Khon Kaen University; and Dr. Wilaiphorn Thinkhamrop, Data Management and Statistical Analysis Center (DAMASAC), Faculty of Public Health, Khon Kaen University for their statistical support.

## Author Contributions

**Conceptualization:** Pisey Vong, Pannee Banchonhattakit, Chamroen Pall.

**Data curation:** Pisey Vong, Samphors Sim.

**Formal analysis:** Pisey Vong.

**Investigation:** Samphors Sim.

**Methodology:** Pisey Vong, Pannee Banchonhattakit, Samphors Sim, Chamroen Pall.

**Project administration:** Pisey Vong.

**Software:** Pisey Vong.

**Supervision:** Pisey Vong, Pannee Banchonhattakit.

**Validation:** Pisey Vong.

**Visualization:** Pisey Vong, Pannee Banchonhattakit, Chamroen Pall, Rebecca S. Dewey.

**Writing – original draft:** Pisey Vong, Samphors Sim, Rebecca S. Dewey.

**Writing – review & editing:** Pisey Vong, Rebecca S. Dewey.

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
