## [Decision Letter · Decision Letter 0]

10 Nov 2020

PONE-D-20-17725

Unhygienic stool-disposal practices among children under five in Cambodia: evidence from a demographic and health survey

PLOS ONE

Dear Dr. Vong,

Thank you for submitting your manuscript to PLOS ONE. After careful consideration, we feel that it has merit but does not fully meet PLOS ONE’s publication criteria as it currently stands. Therefore, we invite you to submit a revised version of the manuscript that addresses the points raised during the review process.

The authors have provided an interesting and important analysis. Please address the reviewers' concerns in order to improve the manuscript for potential publication. All of the below comments should be addressed and please also consider the comments of the third reviewer which were made directly in text comments on the manuscript. When addressing the comments, please indicate where the text was changed in the manuscript in response to each comment. Thank you.

1- I feel sometime confusing to the title of the manuscript as you mention "... practices among children under five...". If possible, I would suggest you to add "... practices among mother/ care giver of children under five years..." to make it more specific and not confusing to reader.

2- Is affiliation of corresponding author in Pursat province or Kampong Cham province? I found this confusing, please make this consistent and should not duplicated.

3- In the "Ethics" section you mention that you obtained data from CDHS, can you add the ref. number of the approval letter?  

4- As in above section, can you please clarify why this analysis approved by Khon Kean University but not National Ethics Committee for Health Research, Ministry of Health to Cambodia?

5- I think you may add in text citation and reference to your forth sentence in the introduction section.

6- In the second paragraph of the introduction, you abbreviated NGOs and put the full phrase in bracket. I think you may write the full phrase first then abbreviated it in bracket to make it consistent to other abbreviation of other phrases.

7- You may add "year" after "age 5" in the second line of the "Explanatory variables" section.

8- You may add in text citation and reference to the first sentence of "Statistical analysis" section.

9- I think the first paragraph of your "result" section, I think you may start some sentences with letter rather than number.

10- I think you specify subheading in the result section without term bivariate and multivariate analysis.

Reviewer #3: Here is a list of specific comments. Note: page numbering in reviews and comments is based on ruler applied in Editorial Manager-generated PDF. Line numbering is not available.

1. Page 1, Methods, line 2: I suggest replacing “multivariate” with ‘multivariable’ throughout the manuscript.

2. Page 1, Methods, lines 3: I suggest replacing “the Stata command svyset” with ‘Stata’ because the svyset command only declared the design but did not perform the analyses.

3. Page 1, Results: I suggest including more numeric associations. Maybe consider to list the most important factors because of the word limits.

4. Page 3, Data Source, lines 4–6: Were the seven data files “fertility”, . . . , and “nutrition” that were mentioned in the previous sentence? If so, I suggest replacing “household data” and “chilren’s data” with the names introduced previously.

5. Page 3, Data Source, lines 6–8: The sentence, “these data were used ... using multiple logistic regression”, was irrelevant for the Data Source section. I suggest removing it.

6. Page 4, Explanatory Variables, lines 10–15: Can these two sentences, “mother’s . . . ” and “similarly, the father’s . . . ”, be combined as one sentence, ‘mother’s and father’s education . . . ’?

7. Page 4, Explanatory Variables, lines 15–17: I was curious why child’s age and mother’s age were treated differently.

8. Page 5, Statistical Analysis, lines 1–2: The sentence, “the women’s individual sample weightings ...”, seemed duplicated comparing with the third subsequent sentence, “cross-tabulations ...”. If they were different, I suggest clarifying their differences. Otherwise, I suggest combining the two sentences into one.

9. Page 5, Statistical Analysis, lines 6–7: I was not sure the implication of “bivariate and multivariate”. Did it mean ‘unadjusted and adjusted’?

10. Page 5, Statistical Analysis, lines 7–9: The outcome was binary. The F-test and linearity test seemed inappropriate for the binary outcome. Unadjusted logistic regressions can be used to examine associations between the binary outcome and explanatory variables. In addition, please elaborate the linearity test.

11. Page 6, Statistical Analysis, line 2: Please make it clear that “the initial model” referred to “the multivariable binary logistic regression” in the second subsequent sentence.

12. Page 6, Statistical Analysis, lines 2–4: Shouldn’t the multi-collinearity exist among explanatory variables, not the outcome variables? Please clarify and elaborate further. I did not find any findings regarding the multi-collinearity in the Results section.

13. Page 6, Statistical Analysis, line 5: I thought the svyset command was only used to declare the survey data. Shouldn’t be there more commands used for further analyses?

14. Page 8, Table 1: I suggest including weighted numbers and percentages along with the unweighted ones.

15. Page 9, Table 2: I suspected the numbers in the COR column were unadjusted odds ratios. If so, I suggest replacing “COR” with ‘odds ratio’ and replacing “bivariate analysis’ with ‘unadjusted binary logistic regression’.

We look forward to receiving your revised manuscript.

Kind regards,

Alessandra N. Bazzano

Academic Editor

PLOS ONE

Journal Requirements:

2. Please provide a copy of the questionnaire used in the study as a supplemental file. We also ask that you amend your Ethics Statement and Methods section to specify whether informed participant consent was provided and if so, what type of consent (i.e., written, verbal, etc).

Reviewers' comments:

Reviewer's Responses to Questions

**Comments to the Author**

1. Is the manuscript technically sound, and do the data support the conclusions?

Reviewer #1: Yes

Reviewer #2: Yes

Reviewer #3: Partly

2. Has the statistical analysis been performed appropriately and rigorously? 

Reviewer #1: Yes

Reviewer #2: Yes

Reviewer #3: No

3. Have the authors made all data underlying the findings in their manuscript fully available?

Reviewer #1: Yes

Reviewer #2: Yes

Reviewer #3: No

4. Is the manuscript presented in an intelligible fashion and written in standard English?

Reviewer #1: Yes

Reviewer #2: Yes

Reviewer #3: No

5. Review Comments to the Author

Reviewer #1: Dear Authors,

Many thanks for your submission to publish this paper. I found the paper is very interesting and novel in this topic to Cambodia. However, I have some minor comments for your consideration to improve your manuscript as below.

1- I feel sometime confusing to the title of the manuscript as you mention "... practices among children under five...". If possible, I would suggest you to add "... practices among mother/ care giver of children under five years..." to make it more specific and not confusing to reader.

2- Is affiliation of corresponding author in Pursat province or Kampong Cham province? I found this confusing, please make this consistent and should not duplicated.

3- In the "Ethics" section you mention that you obtained data from CDHS, can you add the ref. number of the approval letter?

4- As in above section, can you please clarify why this analysis approved by Khon Kean University but not National Ethics Committee for Health Research, Ministry of Health to Cambodia?

5- I think you may add in text citation and reference to your forth sentence in the introduction section.

6- In the second paragraph of the introduction, you abbreviated NGOs and put the full phrase in bracket. I think you may write the full phrase first then abbreviated it in bracket to make it consistent to other abbreviation of other phrases.

7- You may add "year" after "age 5" in the second line of the "Explanatory variables" section.

8- You may add in text citation and reference to the first sentence of "Statistical analysis" section.

9- I think the first paragraph of your "result" section, I think you may start some sentences with letter rather than number.

10- I think you specify subheading in the result section without term bivariate and multivariate analysis.

Thank again for your consideration to revise before publication.

All the best,

Reviewer #2: The article is all about the Unhygienic stool-disposal practices among children under five in Cambodia. the content of the manuscript was extremely informative and reliable. I have some minor comments.

For more I have added some comments in manuscript directly.please see attached

Reviewer #3: Here is a list of specific comments. Note: page numbering in reviews and comments is based on ruler applied in Editorial Manager-generated PDF. Line numbering is not available.

1. Page 1, Methods, line 2: I suggest replacing “multivariate” with ‘multivariable’ throughout the manuscript.

2. Page 1, Methods, lines 3: I suggest replacing “the Stata command svyset” with ‘Stata’ because the svyset command only declared the design but did not perform the analyses.

3. Page 1, Results: I suggest including more numeric associations. Maybe consider to list the most important factors because of the word limits.

4. Page 3, Data Source, lines 4–6: Were the seven data files “fertility”, . . . , and “nutrition” that were mentioned in the previous sentence? If so, I suggest replacing “household data” and “chilren’s data” with the names introduced previously.

5. Page 3, Data Source, lines 6–8: The sentence, “these data were used ... using multiple logistic regression”, was irrelevant for the Data Source section. I suggest removing it.

6. Page 4, Explanatory Variables, lines 10–15: Can these two sentences, “mother’s . . . ” and “similarly, the father’s . . . ”, be combined as one sentence, ‘mother’s and father’s education . . . ’?

7. Page 4, Explanatory Variables, lines 15–17: I was curious why child’s age and mother’s age were treated differently.

8. Page 5, Statistical Analysis, lines 1–2: The sentence, “the women’s individual sample weightings ...”, seemed duplicated comparing with the third subsequent sentence, “cross-tabulations ...”. If they were different, I suggest clarifying their differences. Otherwise, I suggest combining the two sentences into one.

9. Page 5, Statistical Analysis, lines 6–7: I was not sure the implication of “bivariate and multivariate”. Did it mean ‘unadjusted and adjusted’?

10. Page 5, Statistical Analysis, lines 7–9: The outcome was binary. The F-test and linearity test seemed inappropriate for the binary outcome. Unadjusted logistic regressions can be used to examine associations between the binary outcome and explanatory variables. In addition, please elaborate the linearity test.

11. Page 6, Statistical Analysis, line 2: Please make it clear that “the initial model” referred to “the multivariable binary logistic regression” in the second subsequent sentence.

12. Page 6, Statistical Analysis, lines 2–4: Shouldn’t the multi-collinearity exist among explanatory variables, not the outcome variables? Please clarify and elaborate further. I did not find any findings regarding the multi-collinearity in the Results section.

13. Page 6, Statistical Analysis, line 5: I thought the svyset command was only used to declare the survey data. Shouldn’t be there more commands used for further analyses?

14. Page 8, Table 1: I suggest including weighted numbers and percentages along with the unweighted ones.

15. Page 9, Table 2: I suspected the numbers in the COR column were unadjusted odds ratios. If so, I suggest replacing “COR” with ‘odds ratio’ and replacing “bivariate analysis’ with ‘unadjusted binary logistic regression’.

6. PLOS authors have the option to publish the peer review history of their article (what does this mean?). If published, this will include your full peer review and any attached files.

Reviewer #1: No

Reviewer #2: No

Reviewer #3: No

---

## [Author Response · Author response to Decision Letter 0]

21 Dec 2020

Taken care. Please, see the attached files for each Respond to Reviewers (Reviewer #1, #2 and #3)

---

## [Decision Letter · Decision Letter 1]

26 Feb 2021

PONE-D-20-17725R1

Unhygienic stool-disposal practices among mothers of children under five in Cambodia: evidence from a demographic and health survey

PLOS ONE

Dear Dr. Vong,

Thank you for submitting your manuscript to PLOS ONE. After careful consideration, we feel that it has merit but does not fully meet PLOS ONE’s publication criteria as it currently stands. Therefore, we invite you to submit a revised version of the manuscript that addresses the points raised during the review process.

ACADEMIC EDITOR:

Thank you for addressing the reviewer comments from the previous version. Please finalize revision in response to the comment of reviewer:

" Many thanks for taken care of all my comments. I appreciate your response to all comments. However, in my comment #4, I would advice adding back the sentence of additional ethics approval from your university and you may add a phrase of "As part of my PhD at ...., this study ..."

It is recommended to provide the full detail on ethics approval from all sources.

We look forward to receiving your revised manuscript.

Kind regards,

Alessandra N. Bazzano

Academic Editor

PLOS ONE

Journal Requirements:

Reviewers' comments:

Reviewer's Responses to Questions

**Comments to the Author**

1. If the authors have adequately addressed your comments raised in a previous round of review and you feel that this manuscript is now acceptable for publication, you may indicate that here to bypass the “Comments to the Author” section, enter your conflict of interest statement in the “Confidential to Editor” section, and submit your "Accept" recommendation.

Reviewer #1: All comments have been addressed

Reviewer #3: All comments have been addressed

2. Is the manuscript technically sound, and do the data support the conclusions?

Reviewer #1: Yes

Reviewer #3: Yes

3. Has the statistical analysis been performed appropriately and rigorously? 

Reviewer #1: Yes

Reviewer #3: Yes

4. Have the authors made all data underlying the findings in their manuscript fully available?

Reviewer #1: Yes

Reviewer #3: No

5. Is the manuscript presented in an intelligible fashion and written in standard English?

Reviewer #1: Yes

Reviewer #3: Yes

6. Review Comments to the Author

Reviewer #1: Dear Authors,

Many thanks for taken care of all my comments. I appreciate your response to all comments. However, in my comment #4, I would advice adding back the sentence of additional ethics approval from your university and you may add a phrase of "As part of my PhD at ...., this study ..."

Thanks,

Reviewer #3: (No Response)

7. PLOS authors have the option to publish the peer review history of their article (what does this mean?). If published, this will include your full peer review and any attached files.

Reviewer #1: No

Reviewer #3: No

---

## [Author Response · Author response to Decision Letter 1]

2 Mar 2021

Reply: Taken care: As part of the first author’s PhD at Khon Kaen University, this study was conducted following additional approval received from the Khon Kean University Ethics Committee in Human Research with the ref. number HE632097.

---

## [Editor Report · Decision Letter 2]

4 Mar 2021

PONE-D-20-17725R2

Unhygienic stool-disposal practices among mothers of children under five in Cambodia: evidence from a demographic and health survey

PLOS ONE

Dear Dr. Vong,

Thank you for submitting your manuscript to PLOS ONE. After careful consideration, we feel that it has merit but does not fully meet PLOS ONE’s publication criteria as it currently stands. Therefore, we invite you to submit a revised version of the manuscript that addresses the points raised during the review process.

ACADEMIC EDITOR: Pl

The authors have commendably addressed comments. However please add more information to the sentence on ethics review. Specifically, "Data from the CDHS were used after having obtained permission " please state who the permission was obtained from (Cambodia Ministry ? ICF?). Once that is complete it will be the final revision.

We look forward to receiving your revised manuscript.

Kind regards,

Alessandra N. Bazzano

Academic Editor

PLOS ONE

Journal Requirements:

Additional Editor Comments (if provided):

The authors have commendably addressed comments. However please add more information to the sentence on ethics review. Specifically, "Data from the CDHS were used after having obtained permission " please state who the permission was obtained from (Cambodia Ministry ? ICF?). Once that is complete it will be the final revision.

---

## [Author Response · Author response to Decision Letter 2]

9 Mar 2021

Data from the CDHS were used after having obtained written permission from the ICF

---

## [Editor Report · Decision Letter 3]

10 Mar 2021

Unhygienic stool-disposal practices among mothers of children under five in Cambodia: evidence from a demographic and health survey

PONE-D-20-17725R3

Dear Dr. Vong,

We’re pleased to inform you that your manuscript has been judged scientifically suitable for publication and will be formally accepted for publication once it meets all outstanding technical requirements.

Kind regards,

Alessandra N. Bazzano

Academic Editor

PLOS ONE
---

## [Editor Report · Acceptance letter]

22 Jun 2021

PONE-D-20-17725R3 

Unhygienic stool-disposal practices among mothers of children under five in Cambodia: evidence from a demographic and health survey 

Dear Dr. Vong:

I'm pleased to inform you that your manuscript has been deemed suitable for publication in PLOS ONE. Congratulations! Your manuscript is now with our production department. 

Kind regards, 

on behalf of

Dr. Alessandra N. Bazzano 

Academic Editor

PLOS ONE